# Composite Carbon Foams as an Alternative to the Conventional Biomass-Derived Activated Carbon in Catalytic Application

**DOI:** 10.3390/ma14164540

**Published:** 2021-08-12

**Authors:** Mahitha Udayakumar, Renáta Zsanett Boros, László Farkas, Andrea Simon, Tamás Koós, Máté Leskó, Anett Katalin Leskó, Klara Hernadi, Zoltán Németh

**Affiliations:** 1Advanced Materials and Intelligent Technologies Higher Education and Industrial Cooperation Centre, University of Miskolc, H-3515 Miskolc, Hungary; kemudaya@uni-miskolc.hu; 2Institute of Chemistry, University of Miskolc, H-3515 Miskolc, Hungary; 3Wanhua-Borsod Chem Zrt, Bolyai tér 1, H-3700 Kazincbarcika, Hungary; renata.boros@borsodchem.eu (R.Z.B.); laszlo.farkas@borsodchem.eu (L.F.); 4Institute of Ceramic and Polymer Engineering, University of Miskolc, H-3515 Miskolc, Hungary; simon.andrea@uni-miskolc.hu; 5Institute of Energy and Quality Affairs, University of Miskolc, H-3515 Miskolc, Hungary; koos.tamas@uni-miskolc.hu (T.K.); kkklesko@uni-miskolc.hu (A.K.L.); 6Institute of Mineralogy and Geology, University of Miskolc, H-3515 Miskolc, Hungary; askmate@uni-miskolc.hu; 7Institute of Physical Metallurgy, Metal Forming and Nanotechnology, University of Miskolc, H-3515 Miskolc, Hungary; femhernadi@uni-miskolc.hu; 8Department of Applied and Environmental Chemistry, University of Szeged, H-6720 Szeged, Hungary

**Keywords:** carbon foam, phosgene, catalyst, biomass-derived activated carbon, specific surface area, pore size distribution, metallic impurities

## Abstract

The suitability of a new type of polyurethane-based composite carbon foam for several possible usages is evaluated and reported. A comparison of the properties of the as-prepared carbon foams was performed with widely available commercial biomass-derived activated carbon. Carbon foams were synthesized from polyurethane foams with different graphite contents through one-step activation using CO_2_. In this work, a carbon catalyst was synthesized with a moderately active surface (S_BET_ = 554 m^2^/g), a thermal conductivity of 0.09 W/mK, and a minimum metal ion content of 0.2 wt%, which can be recommended for phosgene production. The composite carbon foams exhibited better thermal stability, as there is a very little weight loss at temperatures below 500 °C, and weight loss is slower at temperatures above 500 °C (phosgene synthesis: 550–700 °C). Owing to the good surface and thermal properties and the negligible metallic impurities, composite carbon foam produced from polyurethane foams are the best alternative to the conventional coconut-based activated carbon catalyst used in phosgene gas production.

## 1. Introduction

Carbon foams are 3D-structured porous materials that have been attracting increasing attention due to their versatile properties and wide range of applications. Carbon foams can be made through blowing carbon precursors, the template replica method, the compression of exfoliated graphene, and the assembly of graphene nanosheets [1]. Template carbonization is the simplest method, and uses foam templates such as polyurethane (PU), melamine and zeolite foams [2,3,4]. By choosing a suitable template, porous materials can be synthesized with defined porosity, surface area, particle size, and crystallinity [5]. Carbon foams are used extensively as electrodes in energy storage devices, for heavy metal and organic dye removal, and for CO_2_ adsorption [6,7,8,9]. Additionally, 3D-structured carbon materials have potential applications in catalytic chemical reactions due to their large geometric surface areas and interconnected pores, which provide well-defined pathways allowing reactants to easily access active sites. Carbon foams can simplify the catalytic process and reduce the costs related to packed-bed or slurry catalytic material [10]. However, the catalytic application of 3D carbon foams remains little explored in many chemical reactions.

Mostly, hydrogenation processes have been studied using carbon foams, such as the gas-phase hydrogenation of acetone to 2-propanol and methyl isobutyl ketone on activated and Ni-decorated carbon foam [11] and the hydrogenation of D-xylose to D-xylitol on ruthenium-decorated carbon foam [12]. Additionally, Bukhanko et al. [13] studied the gas-phase synthesis of isopropyl chloride from isopropanol and HCl over alumina and flexible 3D carbon foam-supported catalysts. In our recent work, we developed and tested a carbon foam-supported Pd catalyst for the hydrogenation of nitrobenzene as a model reaction of nitro compound hydrogenation [14]. On the other hand, no previous works have recommended or investigated the suitability of carbon foams as catalysts in industrially important phosgene production processes.

Phosgene (carbonyl chloride, COCl_2_) is an important industrial chemical used in the manufacture of polyurethanes, polycarbonates, pharmaceuticals and agrochemicals [15]. Industrially, phosgene gas is produced by the gas-phase reaction of purified carbon monoxide with chlorine over a porous activated carbon (AC) catalyst in multitubular reactors [16]. Activated carbon with good adsorptive capacity is suitable for use as a catalyst. However, this process has various problems, such as the formation of unwanted by-products (carbon tetrachloride and carbon dioxide) and the low efficiency of production. Since phosgene formation is highly exothermic (ΔH = −107.6 kJ/mol), the reaction temperature in the middle of the catalyst bed can reach as high as 500 °C or more [15]. Above 200 °C, the phosgene formed tends to dissociate, and hence the reactor must be equipped with a proper cooling system [17]. High-quality phosgene is primarily required for use in high-end products. The selection of suitable catalysts for use is the most significant aspect of phosgene synthesis. In 1951, Potter and Baron [18] studied the reaction kinetics of CO and Cl_2_ at temperatures between 25 and 80 °C over a granular carbon catalyst bed. Later, Abrams et al. [19] reported that the usage of new synthetic carbon material reduced the carbon tetrachloride level by an order of magnitude and improved the lifetime of the catalyst to 5 to 10 times that obtained using conventional coconut-based carbons. Mitchell and co-workers [15] evaluated the suitability of eight commercial-grade biomass-derived activated carbon catalysts for phosgene synthesis with respect to their relative reactivity and oxidative stability.

When commercial biomass-derived activated carbon catalyst is used without treatment, a large number of impurities—especially carbon tetrachloride—are formed in the obtained reaction gas as by-products as a result of impurities in the activated carbon or due to the increase in the reaction temperature generally caused by the rapid reaction [20]. Lowering the content of such metallic impurities to a specific value by washing the catalyst with acid is considered to be an effective solution to the problem [20]. However, treating the large volumes of catalyst needed for the mass production of phosgene is troublesome and could cause several bottlenecks. If catalysts of high purity are employed, excellent and economical catalyst efficiency can be obtained. Other than using a biomass-derived carbon catalyst, Ajmera et al. [21] presented a silicon-based packed-bed reactor for the heterogeneous gas-phase production of phosgene. Furthermore, Gupta et al. incorporated the fullerene (C_60_) [22] and nitrogen-modified carbon nanomaterials derived from N-modified polymer material [23] as catalysts for the formation of phosgene gas.

From an industrial point of view, carbon catalyst produced by means of a simple process with moderately active surfaces, minimum metal ion content, and relatively good thermal stability and conductivity would be ideal for phosgene production. In this work, we evaluate and report the suitability of graphite-activated carbon composite foam catalysts for the production of phosgene gas. Graphite is considered a good heat conductor in fixed-bed reactors for cooling systems using chemical adsorption, and its thermophysical properties are more favourable in this study. The graphite is added as a filler material to improve the thermal stability and conductivity of the turbostratic carbon foam. A comparison of the properties of the as-prepared carbon foams was performed with widely available commercial coconut-based activated carbon. Carbon foams were synthesized from the polyurethane foams with different graphite content through a simple one-step activation process using CO_2_. We tested the morphological, textural and thermal properties of the as-prepared composite carbon foams. Owing to the good surface and thermal properties and negligible metallic impurities, carbon foams produced from polyurethane foams present the best alternative to the conventional biomass-derived activated carbon catalyst used in phosgene production.

## 2. Materials and Methods

The composite activated carbon foams (ACFs) were synthesized from polyurethane foams containing different weight percentages of graphite (10%, 20% and 30%) (received from BorsodChem Zrt., Kazincbarcika, Hungary) in a similar manner to that reported in our recent work [8]. A flow diagram of the synthesis process is given in Figure 1. In brief, the precursor solution used for impregnation of the PU foam was 2.5 g/mL concentrated sucrose (Magyar Cukor Zrt., Koronás TM, Kaposvár, Hungary) solution prepared from 2.8% (*v*/*v*) dilute sulphuric acid solution (VWR International Kft., Debrecen, Hungary). The foams were immersed and kept in the precursor solution for 12 h followed by atmospheric drying overnight and oven-drying at 110 °C for 10 h. The impregnated foams were directly activated in a CO_2_ atmosphere (Messer Group GmbH, Siegen, Germany) with a flow rate of 200 mL/min at 1000 °C at a heating rate of 10 °C/min with a dwell time of 100 min in a tube furnace (Carbolite^®^ 1200 °C Split Tube furnace VST 12/900, Carbolite Ltd., Neuhausen, Enzkreis, Germany). The simple activated carbon foam is denoted as ACF and the composite carbon foams are denoted as ACF-G10, ACF-G20 and ACF-G30, respectively, on the basis of their graphite content. The properties of the as-prepared carbon foams were compared with a commercial coconut-based activated carbon catalyst (named ‘AC-X’) to determine their suitability as catalysts in the phosgene formation process.

The surface morphology of the catalysts was investigated by scanning electron microscopy (SEM) (Thermo Scientific Helios G4 PFIB Cxe, Waltham, MA, USA) and the elemental analysis of the samples was performed by energy-dispersive X-ray spectroscopy (EDX) measurement using the scanning electron microscope and a Röntec XFlash Detector 3001 SDD (Billerica, MA, USA) device.

Quantitative determination of metallic impurities in the catalysts was conducted using a Varian 720 ES inductively coupled plasma optical emission spectrometer (ICP-OES, SpectraLab, Markham, ON, Canada) using the Merck Certipur ICP multi-element standard IV. For ICP-OES analysis, a pre-treatment method of dry ashing coupled with acid extraction was used: 2 g samples were placed in a crucible and ashed in a muffle furnace at 900 °C for 3 h. After cooling, the crucibles were washed with 6 mL of 2.0 wt% diluted nitric acid and heated on a hot plate for 20 min at 110 °C. The extraction solutions were diluted up to 25 mL with ultrapure water and then analyzed by ICP-OES.

Nitrogen adsorption–desorption experiments were carried out at 77 K to determine the Brunauer–Emmett–Teller [24] (BET) specific surface area using an ASAP 2020 instrument (Micromeritics Instrument Corp. Norcross, GA, USA). Five different data points in the P/P_0_ range 0 to 0.3 were used to determine the BET surface area. Before each measurement, the samples were degassed by holding at 90 °C for 24 h. The total pore volume was calculated from N_2_ adsorption data at a relative pressure of 0.97. The external surface area and micropore volume were obtained using the t-plot method. The micropore size distribution was determined by the micropore analysis MP method and the BJH model was applied to show the mesopore size distribution.

Thermogravimetric analysis (TGA) was carried out in a ceramic (corundum) crucible using a Derivatograph C/PC instrument (MOM Szerviz Kft., Budapest, Hungary) to measure the thermal behavior of the ACFs from 25 to 1200 °C at a heating rate of 10 °C/min in the oxygen atmosphere.

The thermal conductivity of the catalysts was measured at 25 and 180 °C using a C-Therm TCi Thermal Conductivity Analyzer (TH91-13-0082), (Thermophysical Instruments, Berlin Germany) applying the modified transient plane source (MTPS) method. For each composition and temperature, five measurements were performed and averaged on the powdered samples.

## 3. Results and Discussion

The properties of the as-prepared carbon foams were tested, and their suitability as catalysts in the production of phosgene gas is discussed in the following sub-sections.

### 3.1. Surface Morphology

The surface morphologies of the carbon catalysts were analyzed using a scanning electron microscope; the micrographs are shown in Figure 2. The ACFs exhibited a disordered microstructure with pores and an interconnected thread-like fibrous structure, as shown in Figure 2a–c. The ACFs are more porous at the same magnification and are different from the commercial activated carbon AC-X. The porous structure of ACFs facilitates the transportation of the reactants chlorine and carbon monoxide molecules inside the catalyst bed, which has a significant influence on the adsorption kinetics and diffusion rate of the reactants within the catalyst.

### 3.2. Elemental Analysis

EDX analysis was performed for the composite carbon foams and commercial activated carbon catalysts to establish their elemental compositions. The EDX spectra are presented in Figure 3. The spectra of the ACFs (Figure 3a–c) reveal characteristic peaks of carbon (C) and oxygen (O), while in the spectrum of AC-X (Figure 3d), in addition to C and O, the characteristic peaks associated with other trace elements such as aluminium (Al), magnesium (Mg), calcium (Ca), etc., are evidenced. The carbon content of AC-X was lower (87.2 wt%), and it contained more impurities (about 4.0 wt%). However, a higher carbon content (>95 wt%) and only a negligible amount of impurities (0.2 wt%) were observed in all cases of composite ACFs. The carbon content might be overestimated using EDX analysis due to the carbonaceous sample holder, but the results are comparable, as all samples were measured under the same conditions. EDX is only able to provide approximate elemental compositions, as the information was obtained from only a small number of locations on the sample surface.

Therefore, to obtain quantitative information regarding the metal ion impurities present in the catalysts, the pretreated samples were analyzed using the ICP-OES technique, and the results are shown in Figure 4. As can be seen from the bar chart (Figure 4), the overall metallic impurities in the commercial activated carbon were 25 to 40 times higher than in the composite foams. This indicates the suitability of the ACFs in phosgene synthesis, as the minimal metal ion content prevents the formation of unwanted by-products such as carbon tetrachloride in the reaction gas.

### 3.3. Textural Properties

The physical characteristics of the catalysts were determined using the nitrogen adsorption test. The BET model was used to estimate the surface area. To determine the micropore volume, the t-plot method was used. Table 1 summarizes the textural properties calculated from the obtained N_2_ adsorption isotherm at 77 K. The composite carbon foams and AC-X catalyst exhibited a typical Type I adsorption isotherm, which is characteristic of microporous materials. The t-plot analyses showed that both types of catalyst possessed significant micropore volumes with only a limited amount of external surface area, whereas the simple ACF displayed a larger external surface area than its micropore area, indicating the mesoporous nature of the carbon foam. The addition of graphite filled and hindered the formation of meso and macropores. However, the ACF with 30% graphite (ACF-G30) had a similar BET specific surface area to that of the commercial activated carbon AC-X, which is commonly used in phosgene production; the BET specific surface area and the total pore volume of the ACFs decreased with increasing graphite content in the raw foams. Thus, it was possible to tune the surface properties of the carbon foams by varying the graphite content, as well as by changing the time and temperature of activation [8], providing a significant advantage. According to the recommendation of IUPAC [25], the BET surface area of a given microporous material estimated by nitrogen adsorption at 77 K is not a realistic probe area, but it represents the apparent surface area, which can be regarded as a useful adsorbent fingerprint, permitting the comparison of the surface properties between the samples.

The pore size distribution of the carbon catalysts is shown in Figure 5. The composite carbon foams exhibited more ultramicropores (<1 nm) (Figure 5a) and slightly fewer mesopores, especially with sizes between 2.5 and 5 nm (Figure 5b), than the AC-X. The ultramicroporosity of the composite carbon foams will not hinder the diffusion of the reactant molecules CO and Cl_2_ on the catalyst surface, as they are smaller than 1 nm (CO—0.113 nm; and Cl_2_—0.199 nm [26]).

Carbon catalysts with surfaces that are not too active (i.e., moderate BET specific surface area) are suitable for phosgene production. The reaction of carbon monoxide and chlorine is vigorous when there are more active sites, resulting in the creation of more hotspots in the reactor and the dissociation of the formed phosgene gas at high temperatures.

### 3.4. Thermal Stability

In a phosgene reactor, high reaction temperature and long-term running result in the carbonization and pulverization of the activated carbon catalyst, resulting in the loss of the most active catalyst material. Therefore, the thermal stability of the carbon catalysts was tested in an oxygen atmosphere, as industrial carbon monoxide contains oxygen impurities; on the one hand, the carbon catalyst is oxidized, while on the other hand, the by-product carbon dioxide is formed. The thermograms of the catalysts are shown in Figure 6. According to the data presented in Figure 6, the composite carbon foams and AC-X differ with respect to their degree of thermal stability. The primary weight loss of AC-X at temperatures up to 180 °C was ascribed to the release of adsorbed water, while the slow thermal decomposition of the carbon structure took place at temperatures between 300 and 600 °C. In a phosgene synthesis reaction, the temperature of the hottest spot of the catalyst bed can reach up to 550–700 °C [27]. The thermograms of the composite carbon foams showed better thermal stability than a simple ACF and the commercial AC-X, with very little weight loss at temperatures below 500 °C and slower weight loss at temperatures above 500 °C.

### 3.5. Thermal Conductivity

Table 2 summarizes the data of the thermal conductivity of the graphite and carbon catalysts measured at 25 °C and 180 °C. During phosgene synthesis, catalysts with poor thermal conductivity insulate hotspots in the reactor, resulting in the unavoidable formation of impurities such as carbon tetrachloride. The commercial activated carbon has a thermal conductivity of 0.11 W/mK at 25 °C, and decreases slightly with increasing temperature. The thermal conductivity of graphite is 0.11 and 0.12 W/mK at room and elevated temperatures, respectively. The suggested activated carbon composite foam catalysts also possess almost the same values as those of the AC-X catalyst, which is commonly used in phosgene synthesis. Although the addition of graphite did not significantly increase the thermal conductivity of the carbon foam, it played a crucial role in tuning the surface properties and thermal stability of the material.

The thermal conductivities of bulk graphite and carbon foams have been reported to be in the range of 52–325 W/mK [28,29] and 151–201 W/mK [30], respectively. However, when materials are measured in powdered form, their thermal conductivities are significantly lower—by factors as high as one to two orders of magnitude—than in bulk form. The effectiveness of heat transfer in powders is strongly dependent on the porosity, packing density, interface contacts, size, shape, and composition of the particles [31]. As can be seen from the SEM micrographs (Figure 2), the surface of the investigated samples possessed a structure characterized by pores and thin fibres. The more complex the surface of the particles is, the lower the thermal conductivity is. All the carbon catalysts possessed significant micropore volume with an average pore size of 1.9–2.0 nm, leading to low values of thermal conductivity.

## 4. Conclusions

The suitability of a new type of polyurethane-based composite carbon foam was evaluated and reported for several possible usages. The introduced graphite-activated carbon composite foam catalyst might be an alternative to the commercial AC catalyst used in the production of phosgene gas. From an industrial point of view, with respect to phosgene production, carbon catalysts with moderately active surfaces, minimal metal ion content, and good thermal stability and conductivity are ideal. The current work focused on the development of such novel catalysts and their characteristics. Carbon foams were synthesized from polyurethane foams with different graphite content through a simple one-step activation process using CO_2_. A comparison of the properties of the as-prepared composite carbon foams with widely available commercial coconut-based activated carbon was performed.

In this work, an optimum BET specific surface area of 554 m^2^/g, a thermal conductivity of 0.09 W/mK, and a metal ion content of 0.2 wt% were achieved, which is ideal for possible industrial applications. Since the phosgene formation reaction is exothermic, the temperature of the reactor rises, causing undesired side reactions catalyzed by the metal ions present in the coconut-based carbon catalyst (about 4.0 wt%). Additionally, the metallic impurities greatly reduce the lifetime of the catalyst. These bottle-necks can be eliminated by using carbon foams. With respect to thermal stability, the ACFs were shown to be a good choice as well, because they suffer from very little weight loss at temperatures below 500 °C, as well as slower weight loss at temperatures above 500 °C (phosgene synthesis: 550–700 °C). Moreover, the surface properties of the carbon foams can be precisely tuned by varying the graphite content, in addition to changing the time and temperature, thus providing the possibility of designing a specific catalyst. Thus, owing to the good surface and thermal properties and the negligible metallic impurities, carbon foam produced from polyurethane foams would be the best alternative to the conventional activated carbon catalysts used in the production of phosgene gas. Since phosgene is a highly toxic gas, in the future it would be desirable to evaluate the catalytic activity and efficiency of ACFs in phosgene formation in a safe and controlled environment.

## Figures and Tables

**Figure 1 materials-14-04540-f001:**
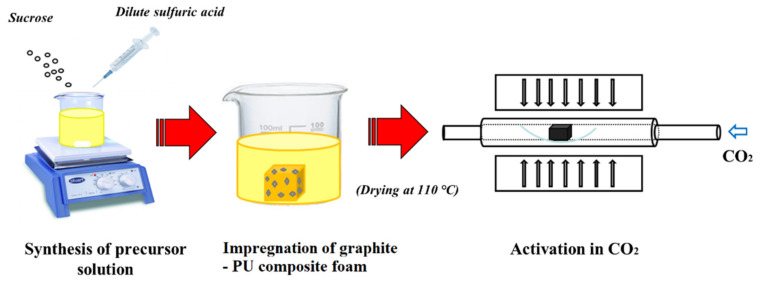
Flow diagram for the synthesis of composite activated carbon foams.

**Figure 2 materials-14-04540-f002:**
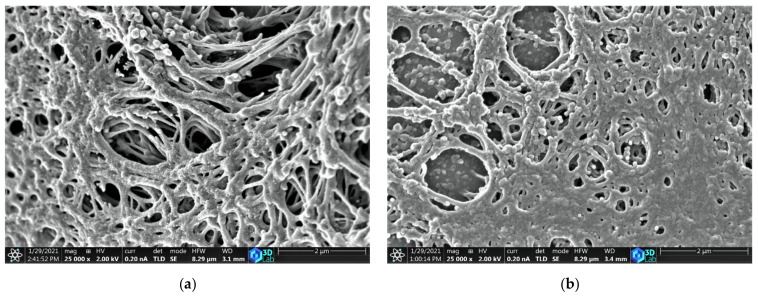
SEM micrographs of (**a**) ACF-G10; (**b**) ACF-G20; (**c**) ACF-G30; and (**d**) AC-X.

**Figure 3 materials-14-04540-f003:**
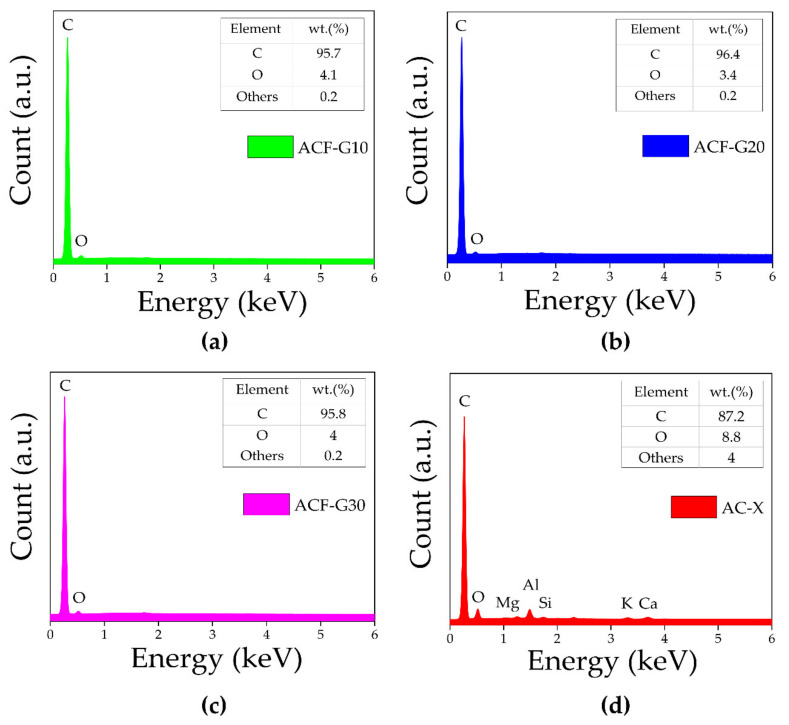
EDX spectra of (**a**) ACF-G10; (**b**) ACF-G20; (**c**) ACF-G30; and (**d**) AC-X.

**Figure 4 materials-14-04540-f004:**
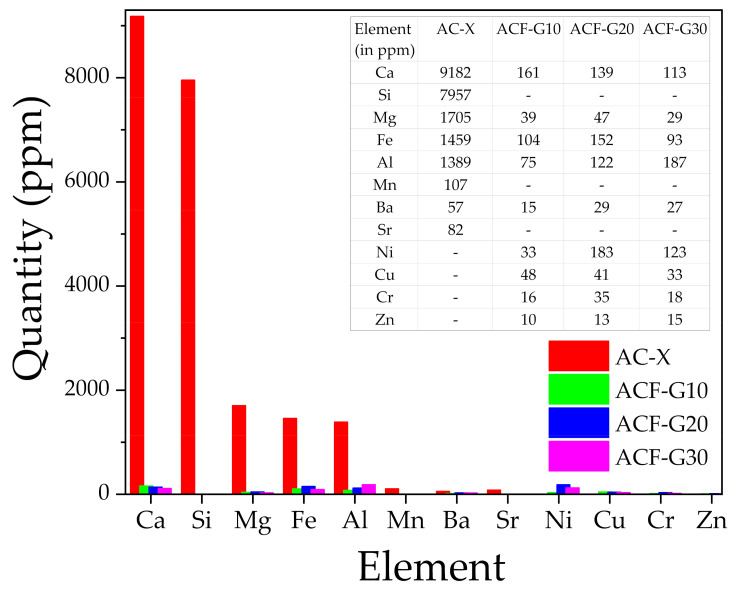
ICP-OES analysis results showing the content of metallic impurities (in ppm) in the catalysts.

**Figure 5 materials-14-04540-f005:**
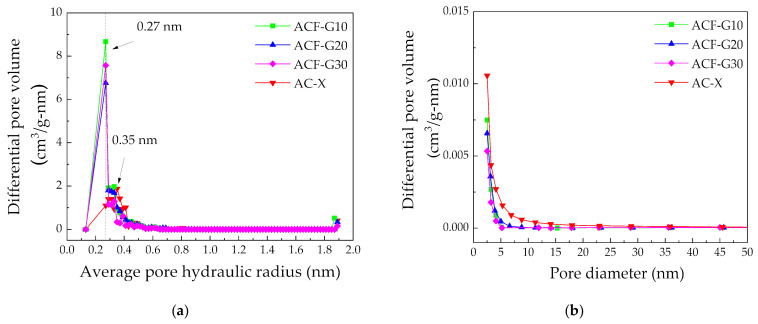
Pore size distribution of the catalysts calculated using the (**a**) MP and (**b**) BJH methods.

**Figure 6 materials-14-04540-f006:**
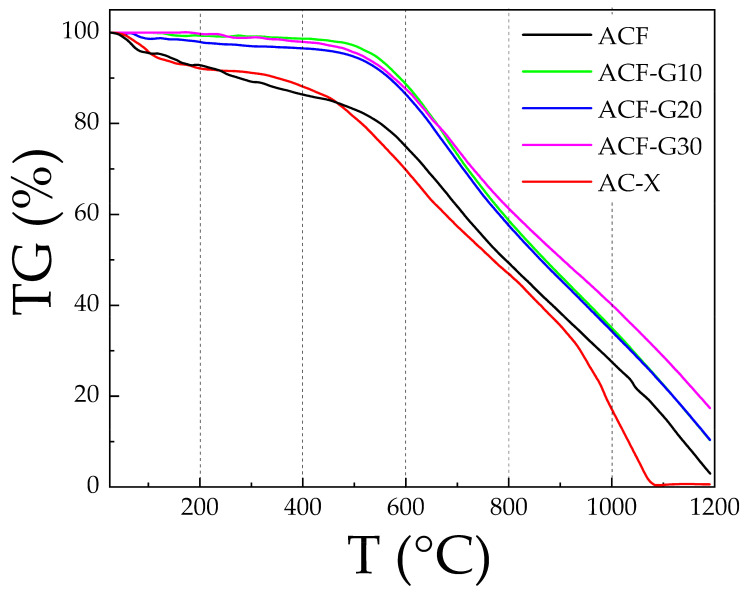
TGA curve of the carbon foams and activated carbon AC-X.

**Table 1 materials-14-04540-t001:** Textural properties of the carbon catalysts.

Sample	BET Surface Area (S_BET_)	Micropore Area	External Surface Area	V_m_ ^a^	V_T_ ^b^	Pore Size
	(m^2^/g)	(m^2^/g)	(m^2^/g)	(cm^3^/g)	(cm^3^/g)	(nm)
ACF	2013	617	1396	0.26	1.00	2.0
ACF-G10	769	690	79	0.32	0.36	1.89
ACF-G20	678	601	77	0.28	0.32	1.89
ACF-G30	554	502	52	0.23	0.26	1.90
AC-X	552	456	96	0.21	0.27	1.98

^a^ Micropore volume determined using t-plot; ^b^ Total pore volume at P/P_0_~0.97.

**Table 2 materials-14-04540-t002:** Thermal conductivity of the carbon catalysts.

Sample	Thermal Conductivity (W/mK)	
	25 °C	180 °C
Graphite	0.11	0.12
ACF	0.08	0.06
ACF-G10	0.08	0.06
ACF-G20	0.09	0.07
ACF-G30	0.09	0.07
AC-X	0.11	0.08

## Data Availability

Data available on request due to restrictions. The data presented in this study are available on request from the corresponding author. The data are not publicly available due to the fact that it is stored on personal computers.

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
