# Peer review of "Composite Carbon Foams as an Alternative to the Conventional Biomass-Derived Activated Carbon in Catalytic Application"

_materials, 2021, doi:10.3390/ma14164540_

Round 1

Reviewer 1 Report

The authors calculate the surface area of microporous materials from nitrogen adsorption isotherm at 77 K; this method is unsuitable for this type of material. IUPAC recommends argon sorption at 87 K (see https://doi.org/10.1515/pac-2014-1117). The authors omit discussion about BET interpretation. I would like to know if they use all points in range P/P0 0-0.3 or use some advanced technique for determining BET region (for example, https://doi.org/10.1016/j.micromeso.2011.05.022).

Thermal analysis was carried out using Netzsch TG209 F3 instrument in an oxygen atmosphere in temperature range RT to 1200 °C. But the temperature range of the instrument is from RT to 1000 °C (see https://www.netzsch-thermal-analysis.com/en/products-solutions/thermogravimetric-analysis/tg-209-f3-tarsus/). Has the device been modified? I´m also missing information on the material of the crucible.

I would also appreciate evaluating the new type of catalyst in terms of estimated costs for industrial use.

Author Response

  1. The authors calculate the surface area of microporous materials from nitrogen adsorption isotherm at 77 K; this method is unsuitable for this type of material. IUPAC recommends argon sorption at 87 K (see https://doi.org/10.1515/pac-2014-1117). The authors omit discussion about BET interpretation. I would like to know if they use all points in range P/P0 0-0.3 or use some advanced technique for determining BET region (for example, https://doi.org/10.1016/j.micromeso.2011.05.022).

Thank you for your valuable comments. Unfortunately, we do not have the chance to measure the samples with argon gas. And, we agree that the surface area determination by N2 adsorption at 77 K using the BET method is not suitable for microporous materials. With the presence of micropores (Type I adsorption isotherm), it is not possible to separate the monolayer and multilayer adsorption and micropore filling (which takes place at very low relative pressure). Besides, the quadrupolar nature of the nitrogen molecule exhibit specific interaction with the surface functional groups of the carbon material – affects not only the orientation of the N2 molecule on the sample but also the micropore filling pressure. However, as mentioned in the IUPAC recommendation, the given BET area determined by N2 is not a realistic probe area, but it represents the apparent surface area, regarded as a useful adsorbent fingerprint (doi.org/10.1515/pac-2014-1117). In this work, the BET areas permit the comparison between the activated carbon foams and commercial activated carbon and also with the literature. It is now included in the article page 7, lines 215-219 as: “Based on the IUPAC recommendation [25], the BET surface area of the given microporous material estimated by nitrogen adsorption at 77 K is not a realistic probe area, but it represents the apparent surface area, regarded as a useful adsorbent fingerprint, which permits the comparison of surface properties between the samples.”

Multiple points (five different data points) were taken between P/P0 0 to 0.3 to determine the BET region (Page 4, lines 144-145: Ideally, five different data points in the P/P0 range 0 to 0.3 were used to determine the BET surface area.). For the determination of micropore volume and external surface area, the deBoer t-plot method was used based on standard isotherms and thickness curves. The extended t-plot method (MP method) extracts micropore volume distribution information from the experimental data.

  1. Thermal analysis was carried out using Netzsch TG209 F3 instrument in an oxygen atmosphere in temperature range RT to 1200 °C. But the temperature range of the instrument is from RT to 1000 °C (see https://www.netzsch-thermal-analysis.com/en/products-solutions/thermogravimetric-analysis/tg-209-f3-tarsus/). Has the device been modified? I´m also missing information on the material of the crucible.

Thank you for the remark. This is an unfortunate mistake and modified the correct instrument name and also included the crucible material on Page 4, lines 151-152 as: “Thermogravimetric analysis (TGA) was carried out in a ceramic (corundum) crucible using a Derivatograph C/PC instrument (MOM Szerviz Kft., Hungary)…”

  1. I would also appreciate evaluating the new type of catalyst in terms of estimated costs for industrial use.

Thank you for the question. If we compare only purchase prices, the use of carbon foam is not economical against the high mass and low cost of activated carbon (of course, AC will be cheaper), but two very important factors need to be considered as advantages of carbon foams:

  1. With the use of carbon foams (less metal contamination, good technical and thermal properties), production is more stable, it does not have to be stopped and it takes a longer time to replace the spent catalyst. The sale of the surplus product overcompensates the difference in the cost of the catalysts.
  2. Carbon foam can be produced from industrial polyurethane waste which creates recycling. This is the biggest advantage.

Moreover, phosgene production with carbon foam will not cause shutdowns (or shutdowns are less) within the year. If we can operate continuously, the profit is higher.

Reviewer 2 Report

The graphite-modified carbon foams were obtained, characterized, and examined for being a catalyst in the synthesis of phosgene in this work.

Page 1, line 22-35

The ‘Abstract’ should contain the most important achievements of the presented work.

‘…a carbon catalyst with moderate active surfaces, minimum metal content, and good thermal properties would be ideal for phosgene production.’ In my opinion, it is too general. The authors should give an appropriate value of an activity, as well as an example of thermal properties there.

            Page 1, line 29

Since the phosgene formation reaction is exothermic….’

This sentence is suitable for the introduction part.

            Page 2, line 64

I can quite agree that phosgene (COCl2) is an important industrial chemical. However, it is highly toxic. Moreover, harmful by-products: carbon tetrachloride and carbon dioxide are evolved, when it is produced. Why don’t you try to eliminate it? May that carbon foam would be useful in the synthesis of dimethyl carbonate, the compound similar in construction but a substrate of Green Chemistry. It has some alternative properties as phosgene, e.g. acts in carbonylation reaction.

            Page 4, line 150

The acronym MP should be explained.

            Page 4, line 161

Although, the ACFs seem to fulfill the requirements of thermal stability and conductivity for the catalyst used in the synthesis of phosgene, the properties of carbon foams should be tested in a real catalytic reaction, not necessary in the phosgene formation, but maybe in another catalytic test, in which carbon samples also recommended for phosgene formation acted well, e. g. commercial AC catalyst.

Author Response

  1. Page 1, line 22-35

The ‘Abstract’ should contain the most important achievements of the presented work.

‘…a carbon catalyst with moderate active surfaces, minimum metal content, and good thermal properties would be ideal for phosgene production.’ In my opinion, it is too general. The authors should give an appropriate value of an activity, as well as an example of thermal properties there.

Thank you for the remark. The abstract has been modified as per the suggestion: Page 1, lines 26-30. “In this work, a carbon catalyst with a moderate active surface (SBET = 554 m2/g), the thermal conductivity of 0.09 W/mK and minimum metal ion content of 0.2 wt% was synthesized, which would be ideal for phosgene production. The composite carbon foams have shown better thermal stability as there is a very little weight loss until 500 ℃ and a slower weight loss at temperatures above 500 ℃ (phosgene synthesis: 550 – 700 ℃).”

  1. Page 1, line 29

Since the phosgene formation reaction is exothermic….’

This sentence is suitable for the introduction part.

Thank you for the remark. This sentence has been removed from the abstract and has already been included in the introduction.

  1. Page 2, line 64

I can quite agree that phosgene (COCl2) is an important industrial chemical. However, it is highly toxic. Moreover, harmful by-products: carbon tetrachloride and carbon dioxide are evolved, when it is produced. Why don’t you try to eliminate it? May that carbon foam would be useful in the synthesis of dimethyl carbonate, the compound similar in construction but a substrate of Green Chemistry. It has some alternative properties as phosgene, e.g. acts in carbonylation reaction.

Thank you for the remark. We discussed in the article that the by-product carbon tetrachloride and carbon dioxide form at high temperatures, which is catalysed by the metal ion impurities present in the catalyst. The biomass-derived activated carbon generally contains high metallic impurities. The formation of such by-products can be controlled by providing a proper cooling system and using a catalyst with better thermal stability and minimum metallic impurities. The suggested carbon foams have shown good thermal stability and very minimal metal ion content, which would reduce or eliminate the formation of CCl4 and CO2 when applied.

Conventional isocyanate production is based on phosgenation due to its effectiveness. Phosgene plays a key role due to its reaction with amine where hydrochloric acid is eliminated as a by-product. This by-product can be further used as raw material. Due to enormous isocyanate production, it will be difficult to replace phosgene with an alternative material because of technological difficulties. Even a small modification has serious risks. On the other hand, the phosgenation step is carried out under controlled circumstances paying high attention to safety. While using dimethyl carbonate, oxidative carbonylation of methylene-dianiline occurs forming carbamate which is then thermally decomposed to methylene diphenyl diisocyante. The reaction occurs at a very high temperature requiring extra energy, moreover, several by-products can be formed as well causing pluggings. The high temperatures might influence the quality of the final product as well. Though dimethyl carbonate is out of our interest, we appreciate your suggestion and try to test it in the future.

  1. Page 4, line 150

The acronym MP should be explained.

MP (extended t-plot) method is the micropore analysis method, to calculate the micropore size distribution using t-plots. This is one of the first methods developed by Mikhail et al. (doi.org/10.1016/0021-9797(68)90271-3) for the analysis of microporous solids. This method uses a t-curve that is constructed by plotting the adsorbed volume of the nitrogen gas versus thickness, t, which represents the average thickness of the adsorbed layer. The t values were calculated as a function of p/p0 for the adsorbed nitrogen gas at 77 K using the Harkins and Jura equation.

  1. Page 4, line 161

Although, the ACFs seem to fulfill the requirements of thermal stability and conductivity for the catalyst used in the synthesis of phosgene, the properties of carbon foams should be tested in a real catalytic reaction, not necessary in the phosgene formation, but maybe in another catalytic test, in which carbon samples also recommended for phosgene formation acted well, e. g. commercial AC catalyst.

Thank you for the remark. We appreciate your recommendation and we have a plan to test the carbon foams in other important catalytic processes in the future.

Round 2

Reviewer 2 Report

line 30, p. 1

'would be ideal...' should be exchanged for 'may be recommended...'

Author Response

Thank you for the remark. The sentence has been modified as suggested (page 1, line 28). (changes in red)